# Reverse Screening of Boronic Acid Derivatives: Analysis of Potential Antiproliferative Effects on a Triple-Negative Breast Cancer Model In Vitro

Miguel Ortiz-Flores [1], Marcos González-Pérez [1], Andrés Portilla [1], Marvin A. Soriano-Ursúa [1], Javier Pérez-Durán [2], Araceli Montoya-Estrada [3], Guillermo Ceballos [1,*] and Nayelli Nájera [1,*]

[1] Sección de Estudios de Posgrado e Investigación, Escuela Superior de Medicina, Instituto Politécnico Nacional, Plan de San Luis y Díaz Mirón s/n, Col. Casco de Santo Tomás, Alc. Miguel Hidalgo, Mexico City 11340, Mexico

[2] Reproductive and Perinatal Health Research Departament, Instituto Nacional de Perinatología "Isidro Espinosa de los Reyes", Montes Urales 800, Alc. Miguel Hidalgo, Mexico City 11000, Mexico

[3] Coordination of Gynecological and Perinatal Endocrinology, Instituto Nacional de Perinatología "Isidro Espinosa de los Reyes", Montes Urales 800, Alc. Miguel Hidalgo, Mexico City 11000, Mexico

\* Correspondence: gceballosr@ipn.mx (G.C.); nnajerag@ipn.mx (N.N.); Tel.: +52-55-57296000 (G.C. & N.N.)

**Abstract:** It has been demonstrated that different organoboron compounds interact with some well-known molecular targets, including serine proteases, transcription factors, receptors, and other important molecules. Several approaches to finding the possible beneficial effects of boronic compounds include various in silico tools. This work aimed to find the most probable targets for five aromatic boronic acid derivatives. In silico servers, SuperPred, PASS-Targets, and Polypharmacology browser 2 (PPB2) suggested that the analyzed compounds have anticancer properties. Based on these results, the antiproliferative effect was evaluated using an in vitro model of triple-negative breast cancer (4T1 cells in culture). It was demonstrated that phenanthren-9-yl boronic acid and 6-hydroxynaphthalen-2-yl boronic acid have cytotoxic properties at sub-micromolar concentrations. In conclusion, using in silico approaches and in vitro analysis, we found two boronic acid derivatives with potential anticancer activity.

**Keywords:** boron; boronic; breast cancer; anticancer drugs; antiproliferative effect

## 1. Introduction

In the last few years, boron-containing compounds (BCC) have gained great relevance due to their potential use in different medicinal chemistry and pharmacology areas [1–3]. The interest is based mainly on their chemical properties, making boron a versatile precursor for chemical reactions due to its reactivity, stability, and low toxicity [4].

Alkyl or aryl boronic acid derivatives are the most common BCC. They have been used as catalysts and precursors in several synthetic reactions [4]; furthermore, these compounds can interact with electron-donating groups, such as nitrogen or oxygen, and abundant atoms of macromolecules, such as receptors, enzymes, or nucleic acids [5]. These characteristics point to boronic acid derivates as compounds with many potential therapeutic applications [5].

Based on these facts, it has been demonstrated over recent years that different BCC, including organoboron compounds, interact with several molecular targets; among these are serine proteases, transcription factors, receptors, and other important molecules [1,5]. Their induced effects have led to clinical trials. Some compounds have been approved by the FDA [6,7], i.e., Bortezomib and Ixazomib, which are proteasome inhibitors used for the treatment of multiple myeloma [6], Tavaborole for onychomycosis, Crisaborole as an inflammation modulator, and Vaborbactam as a β-lactamase inhibitor used in combination with antibiotics for some antimicrobial-resistant agents [6].

Other well-reported and reviewed effects of BCC [1,5] include the inhibition of viral replication, as well as the inhibition of fungal, bacterial, and protozoal growth and reproduction [8], and applications as delivery systems for some drugs [9].

Several approaches to finding the possible beneficial effects of boronic compounds interacting with potential therapeutic targets include in silico tools such as reverse screening [10,11]. This approach can determine the probability of ligand–protein interactions by evaluating the molecule fit inside the binding pocket of a protein target by docking and scoring the key interactions of pharmacophore group(s) in the molecule and the targets [12]. Several servers to perform this process exist, each providing a score that ranks the possible targets based on probability [13].

In this study, the main objective was to find the most probable targets for five aromatic boronic acid derivatives (Figure 1) and evaluate their probable effects as proliferation inhibitors in an in vitro model of a triple-negative type of breast cancer.

(6-hydroxynaphthalen-2-yl) boronic acid (**1**)

[4-(4-propan-2-yloxyphenyl)phenyl] boronic acid (**2**)

pyren-1-yl boronic acid (**3**)

phenanthren-9-yl boronic acid (**4**)

9H-fluoren-2-yl boronic acid (**5**)

**Figure 1.** Chemical structures and IUPAC names of selected boronic acid derivatives. See the methods section regarding the selection procedure.

## 2. Results and Discussion

### 2.1. Prediction of Probable Biological Activity

In searching for the probable biological activity, the Prediction of Activity Spectra for Substances (PASS) online server was employed (see the materials section for details) [14]. Pa (probable activity) and Pi (probable inactivity) are the main outcomes in this analysis. The compounds showing a higher Pa value than Pi are considered possible candidates for biological activity [15,16]. Predicted biological activities are ranked according to the prediction ratio Pa/Pi (Figure 2). Notably, possible actions on peptidyl transferase, 17-β-hydroxysteroid dehydrogenase 1, and estradiol 17-β dehydrogenase were often suggested; they are linked to some breast cancer cell proliferation [17–19].

### 2.2. Therapeutic Target Prediction

To explore the most probable therapeutic targets, SuperPred, PASS-Targets, and Polypharmacology browser 2 (PPB2) were used (see the Section 4 for details). The predictions obtained from the Super-PRED web server [20] for those targets with at least 80% ligand–protein interaction probability are shown in Figure 3. Most of the included proteins were suggested as targets for cancer.

On the other hand, the best 15 predicted targets using the algorithm Extended Connectivity fingerprint ECfp4 NN(ECfp4) + NB(ECfp4) of the Polypharmacology browser 2 server [21] are shown in Table 1.

Similarly, using the PASS-Targets server [22], the potential interactions of the five organoboron compounds with the molecular targets of its database were determined, as shown in Figure 4.

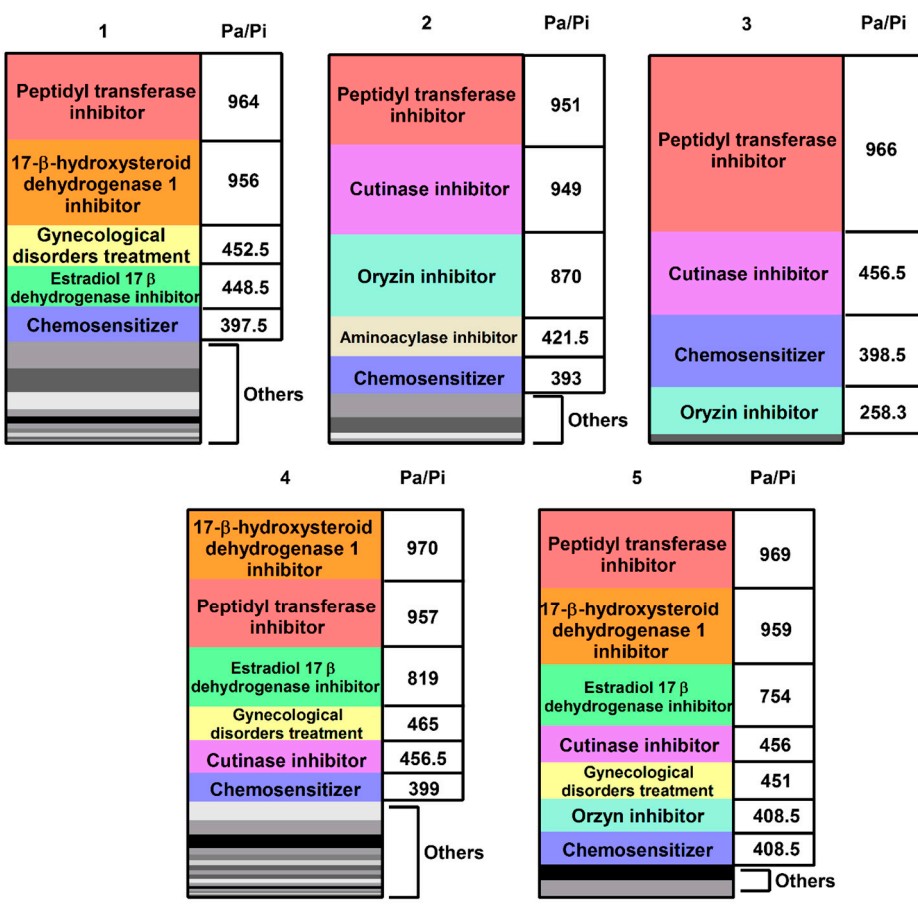

**Figure 2.** Most probable biological activities predicted by PASS online for the five compounds. Activities were ranked according to the ratio (Pa/Pi). Numbers on the panels indicate the corresponding BCC in Figure 1.

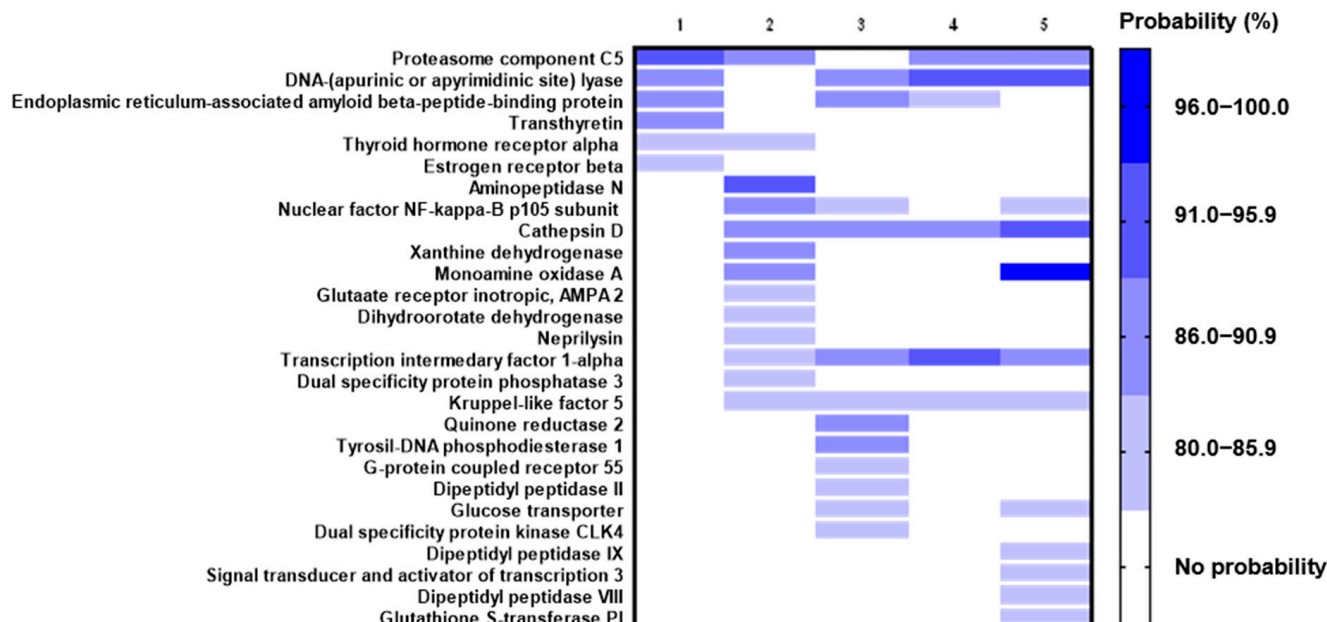

**Figure 3.** Most probable targets for each evaluated compound (probability ≥80%) according to the Super–PRED server. Numbers on the top of the panel indicate the corresponding BCC in Figure 1.

**Table 1.** Best 15 predicted targets, represented as UniProtKB entries (see Table S1 for protein names), for each evaluated compound using Polypharmacology browser 2 servers.

| | Compound | | | |
|---|---|---|---|---|
| 1 | 2 | 3 | 4 | 5 |
| Q92731 | O00763 | P05177 | P22303 | P35462 |
| P11474 | Q13085 | Q07820 | P09917 | P14416 |
| P00918 | P09917 | P31645 | P06276 | P11229 |
| Q8TDS4 | P08588 | Q92731 | P11474 | P21917 |
| P00915 | P45452 | P04818 | Q9BQF6 | Q92731 |
| O43570 | P13945 | P11474 | Q92731 | P06401 |
| Q16790 | P14780 | P08908 | P05177 | P11474 |
| P14061 | P29274 | Q13547 | P08908 | P28335 |
| P37059 | P08253 | P40238 | P27338 | P28223 |
| Q13627 | P37231 | P50406 | P40238 | P35968 |
| P05067 | Q07869 | P22303 | P00918 | P41595 |
| P00533 | Q9Y5N1 | P09874 | Q07820 | P08908 |
| P36888 | P03956 | Q01959 | P09874 | P20309 |
| P11511 | P50281 | P28223 | P31645 | P27338 |
| P08684 | P21453 | Q9UBN7 | P21397 | P35367 |

The numbers of compounds on the top indicate the corresponding BCC in Figure 1.

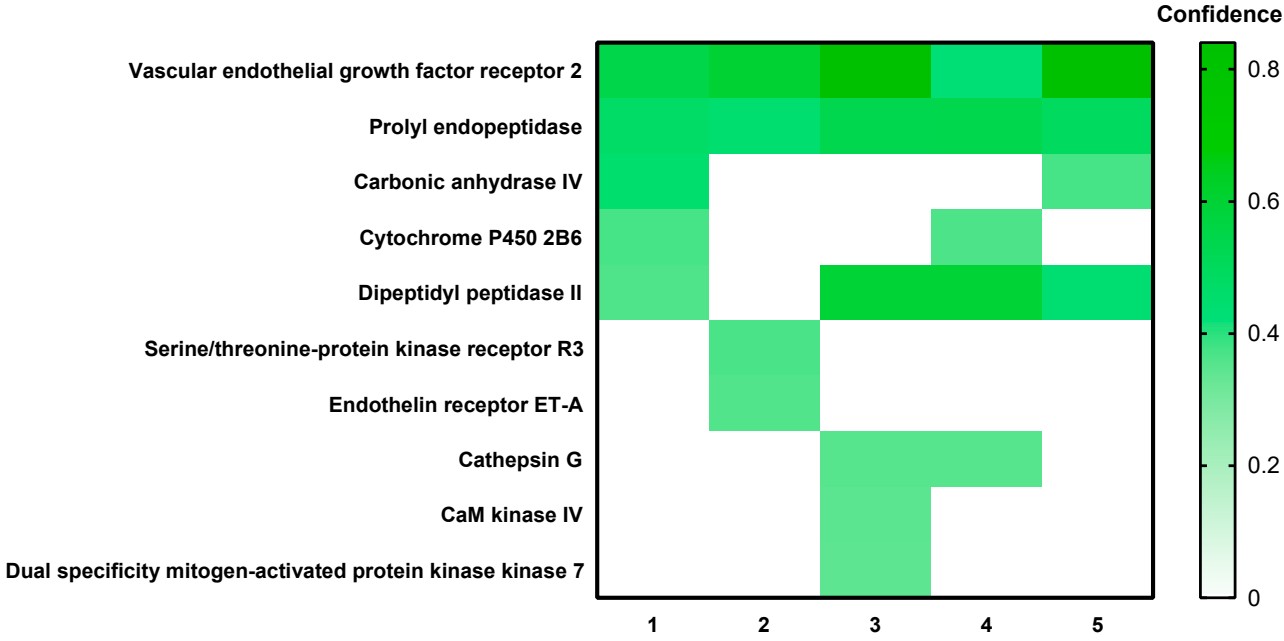

**Figure 4.** Predicted targets with confidence of ≥0.35 for each evaluated compound according to the PASS-Targets server. Numbers below the confidence square indicate the corresponding BCC in Figure 1.

Common targets among compounds and among different servers were used to find associated diseases where the proteins are involved using MalaCards Human Disease Database (see the materials section for details). (Figure 5), and the five best-ranked associated diseases according to the MalaCards database (available at https://www.malacards.org, accessed on 1 November 2022) [23] were obtained and classified (Figure 6).

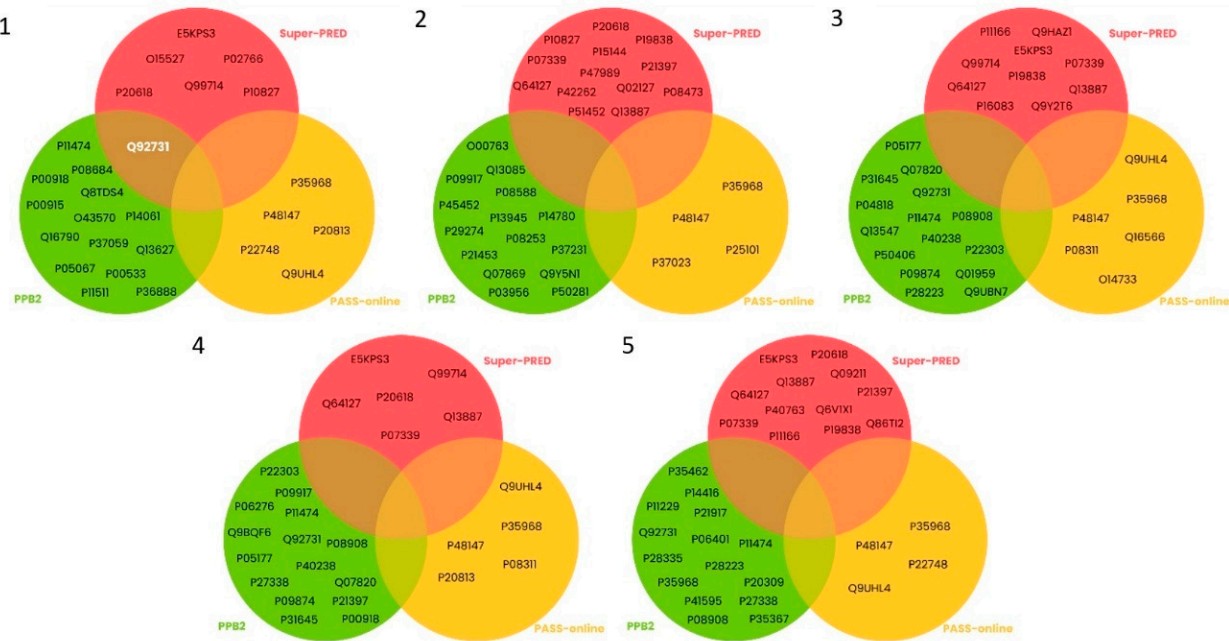

**Figure 5.** Venn diagrams with target prediction comparisons obtained from the three servers. Numbers on each triad indicate the corresponding BCC in Figure 1.

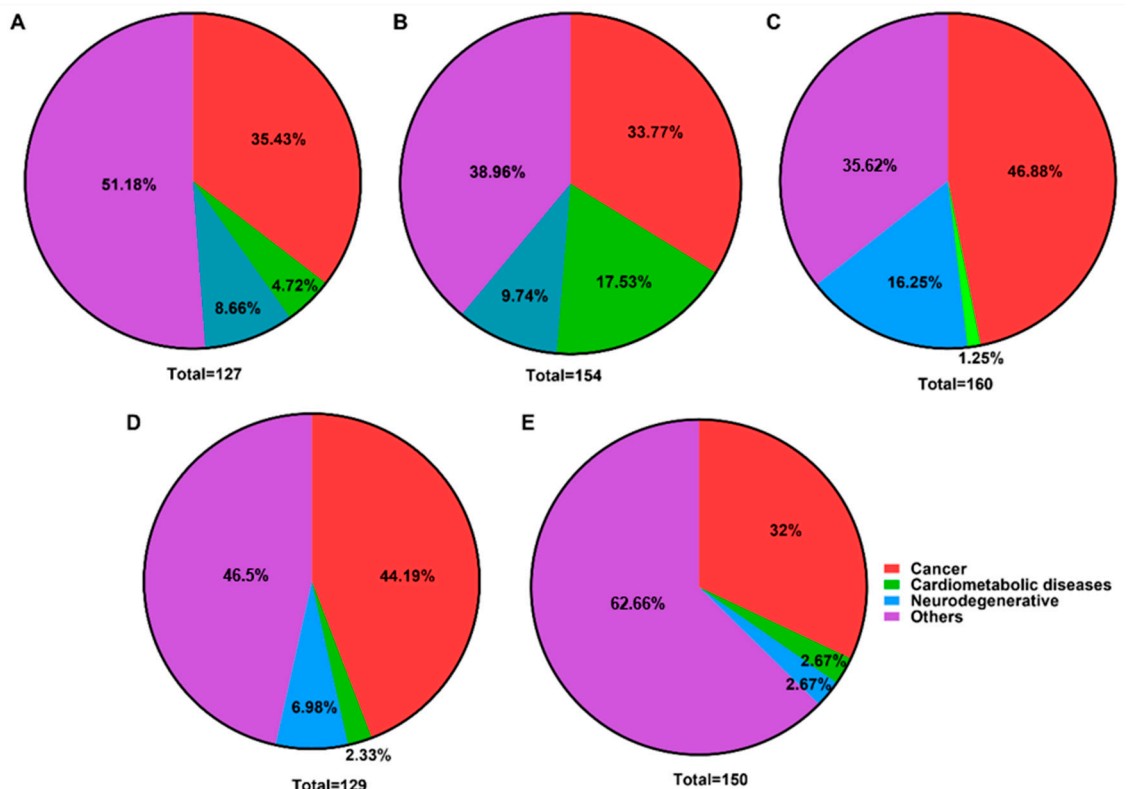

**Figure 6.** Percentages of diseases associated with each therapeutic target of evaluated compounds. Numbers on each pie chart indicate the corresponding BCC in Figure 1.

The results obtained suggested, with high probability, that the main effects of the analyzed molecules could be related to cancer (see the discussion section for details). Considering all predictions, the effects of selected BCC were evaluated in 4T1 cells in culture, a model of triple-negative breast cancer.

### 2.3. T1-Cell Viability Dose–Response Curves

To test the potential antiproliferative effects of the analyzed compounds in 4T1 cells in culture, doxorubicin [0.00001–10 μM] was used as a positive control. A concentration-dependent decrease in cell viability was determined (Figure 7A). On the other hand, the effects of boric acid [0.2–1.2 μM] were analyzed. As expected (since boric acid requires millimolar concentrations to induce a significant disruption in cancer cells [24,25]), the results showed a slight nonsignificant decrease (≅20%) in cell viability (Figure 7B). These results were considered as a negative control.

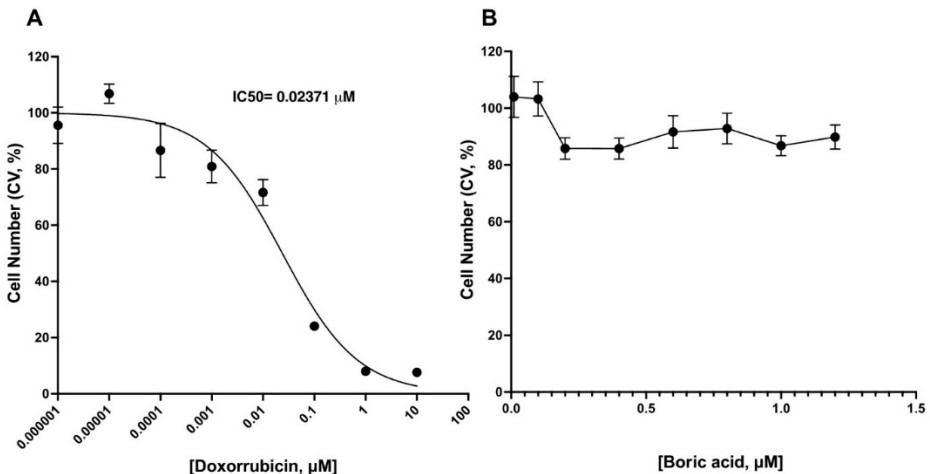

**Figure 7.** The effect of control drugs on the 4T1 cells. (**A**). Doxorubicin (positive control) induced a concentration-dependent decrease in 4T1 cell viability. (**B**). Analysis of the effects of boric acid (negative control) on 4T1 cell viability. Analysis was performed in triplicate. Data are presented as the mean ± standard error (SE).

The effects of 9H-fluoren-2-yl boronic acid (**5**) [0.0001–300 μM] on 4T1 viability were analyzed. Even at higher concentrations than for the other tested BCC, no significant effects of this molecule on cell viability were found (Figure 8A).

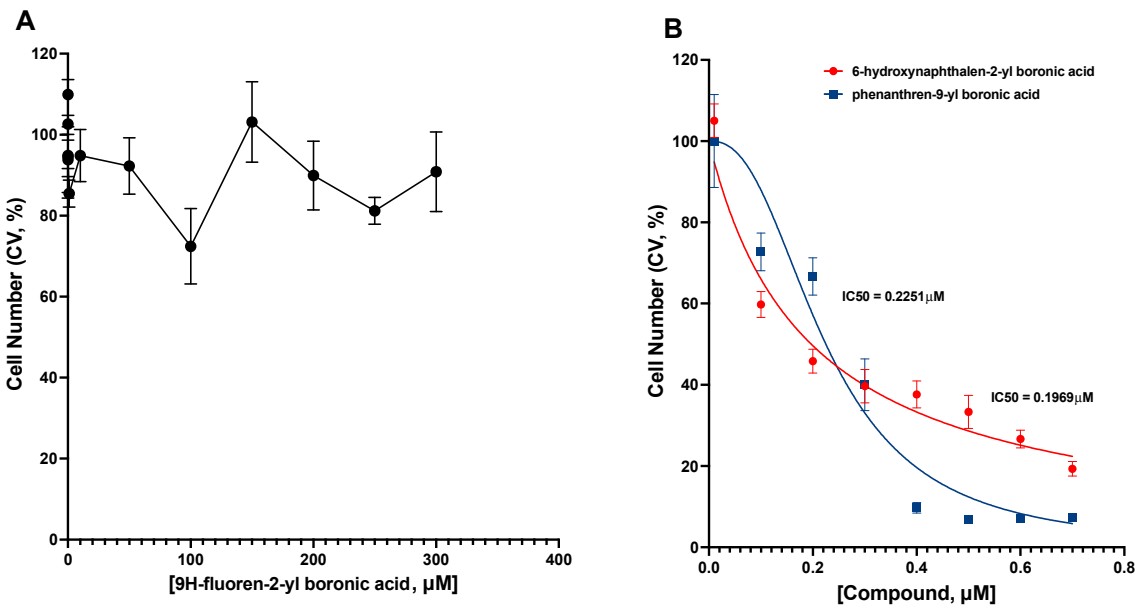

**Figure 8.** The effects of BCC on 4T1 cell viability. (**A**). Effect of compound **1** on cell viability. (**B**). Effects of compound **4** and compound **5** on cell viability. Assays were performed in triplicate. Data are presented as the mean ± SE of the cell viability of 4T1 cells in culture.

The effects of 6-hydroxynaphthalen-2-yl boronic acid (**1**) were analyzed (Figure 8B). A concentration-dependent decrease in cell viability was found, with a half-maximal inhibitory concentration ($IC_{50}$) of 0.1969 µM.

Similarly, phenanthren-9-yl boronic acid (**4**) [0.1–0.7 µM] induced a concentration-dependent decrease in cell viability (Figure 8B), with an $IC_{50}$ value of 0.2251 µM.

The evaluation of the effects of [4-(4-propan-2-yloxyphenyl)phenyl] boronic acid (**2**) and pyren-1-yl boronic acid (**3**) was limited, since these molecules tend to precipitate when they are diluted in RPMI medium (a typical culture medium), making trustable in vitro assays not feasible. A clear relationship was not observed with the water/lipid solubility of these compounds, as compounds **2** and **3** were not predicted as the most water- or lipid-soluble compounds (Table S2).

### 2.4. Effects of BCC Regarding Antiproliferative Effects on Noncancer Cells

After finding that compounds **1** and **4** induced a concentration-dependent decrease in cell number (cytotoxicity) in the cancer cell model in vitro, to examine their effects in a noncancer-related cell type in culture, we chose C2C12 myoblasts for analysis. These cells are of skeletal muscle origin and do not share the cancer-related characteristics of the 4T1 cell type. Compound **1** induced a slight decrease of close to 20% in cell number (Figure 9A), and compound **4** induced a decrease of 28% in cell number (Figure 9B); the results suggest that the effects of these BCC are selectively higher in cancer cells than in noncancer cells.

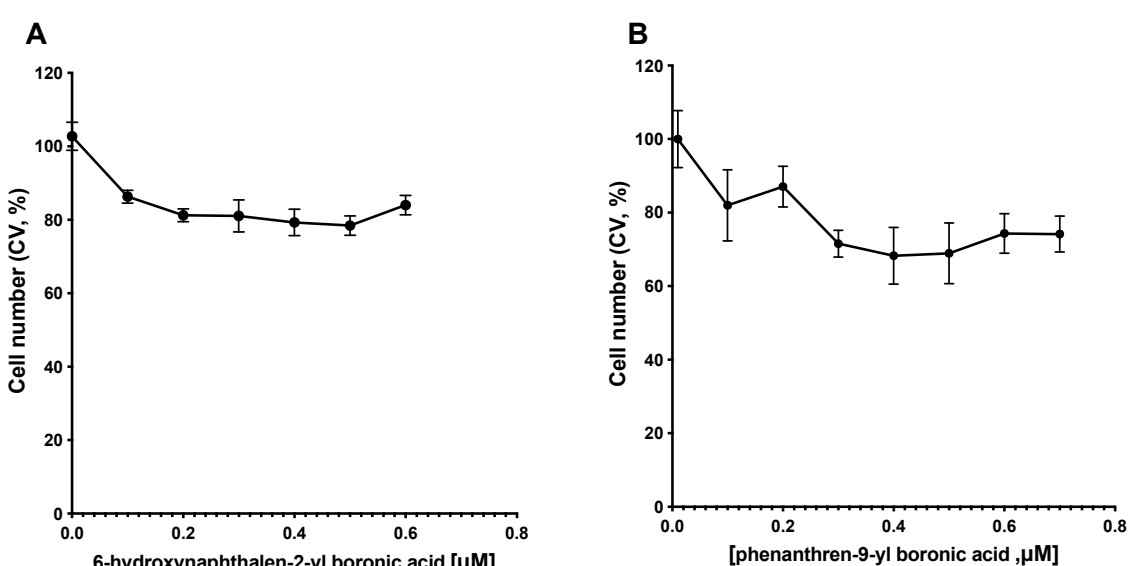

**Figure 9.** Toxicity evaluation in non-cancer cells. (**A**). Effects of 6-hydroxynaphthalen-2-yl boronic acid (**1**) on cell viability in C2C12. (**B**). Effects of phenanthren-9-yl boronic acid (**4**) on myoblasts in culture. Analysis was performed in triplicate. Data are presented as the mean ± SE.

### 3. Discussion

Currently, one of the first approaches to finding the probable biological activity of compounds is through in silico tools [10,26,27]. In this context, reverse screening is one of the most useful methods [10]. In this work, we used this methodology to estimate the most probable therapeutic action of five boronic acid derivatives. Firstly, we utilized the Prediction of Activity Spectra for Substances (PASS) online server [14]; predictions made using this tool are based on the analysis of structure–activity relationships (SAR) in an extensive compound database. The results show the most probable biological activity of compounds ranked according to the active and inactive probabilities ratio (Pa/Pi), meaning that the higher the ratio, the higher the probability of activity. The most frequently predicted activities for all analyzed compounds were as a peptidyl transferase inhibitor and as a chemosensitizer. These results suggest that the five compounds can serve as antibiotics, inhibiting peptide bond formation in bacteria; however, compounds with these

abilities are also involved in cancer progression, and they are studied to develop new anticancer drugs [28,29]. Moreover, these compounds can function as drugs that make tumor cells more sensitive to the effects of chemotherapy [28]. The enzymes related to steroid metabolism, suggested as potential targets, are also considered a target in breast cancer drug design [18,19].

Following the methodological sequence proposed in this work, we predicted the most probable therapeutic targets for each derivative in three online servers. Firstly, according to the Super-PRED server [20], which makes predictions based on machine learning models using the ChEMBL database [30], the probability of binding to 646 human targets, considering 2D similarities, was analyzed. Among the probable targets for each evaluated compound with a binding probability of $\geq$80% were proteasome component C5, DNA-(apurinic or apyrimidinic site) lyase, cathepsin D, transcription intermediary factor 1-$\alpha$, and Kruppel-like factor 5; all these have been explored or suggested as potential targets in certain types of cancer, including breast cancer [17,19].

A second target prediction was then performed using the Polypharmacology browser 2 (PPB2) server [21]. This tool also uses ChEMBL data [30], but it determines ligand similarities using three descriptors (molecular fingerprints encoding composition (MQN), molecular shape and pharmacophores (Xfp), and substructures (ECfp4)), and it uses a combination of mathematical algorithms (nearest neighbor (NN) searches and naïve Bayes (NB) machine learning). Based on previous reports, the Extended Connectivity fingerprint ECfp4 NN(ECfp4) + NB(ECfp4) algorithm was employed since it provides the best recall and precision values compared to other algorithms.

Table 1 shows the first fifteen predicted targets for each compound; four of the compounds (all except for compound **2**) share two targets: estrogen receptor alpha (P11474) and estrogen receptor beta (Q92731). Similarly, compounds **3** and **4** have five targets in common: cytochrome P450 1A2 (P05177), induced myeloid leukemia cell differentiation protein Mcl-1 (Q07820), poly [ADP-ribose] polymerase-1 (P09874), acetylcholinesterase (P22303), and serotonin transporter (P31645). These results suggest that at least four of the tested boronic acid derivatives are highly likely to share a mechanism by which they exert their pharmacological activity.

Finally, a third server to conduct target prediction was utilized: PASS-Targets [22]. This server implements QSAR modeling on data available in the ChEMBL database [30] over 930 human protein targets. The probability of interaction vs. no interaction determines the target scoring (confidence). The higher the confidence, the higher the chance of the positive prediction being true. Considering a confidence of 0.35, the predicted targets for the five organoboron compounds are presented in Figure 4; vascular endothelial growth factor receptor 2 and prolyl endopeptidase were the two common targets in at least four compounds. Both are also considered breast cancer targets [31,32].

After the analysis, targets predicted in common were depicted in Venn diagrams. Two proteins were targets on the three servers: estrogen receptor beta and vascular endothelial growth factor receptor 2—both intimately related to neoplastic pathology growth [18,32].

After the target prediction, the MalaCards database (a database of 22,091 human diseases, modeled in gene cards) [23] was searched for the top five associated diseases (Figure 5). This approach allows us to classify diseases according to (1) the reliability of evaluation and (2) the diseases with the greatest morbi-mortality among the world's population. The results suggested that cancer, cardiometabolic diseases, neurodegenerative diseases, and others (including those diseases for which pharmacological evaluation in vitro or in vivo is complicated, as well as less frequent diseases or those with a favorable survival outlook with adequate treatment, such as rheumatoid arthritis, anxiety, osteoporosis, etc.) are the most probable therapeutic targets for the selected boronic compounds.

The merged results from the all three servers show that approximately 32–46% of the protein targets are associated with cancer, which is the second most abundant subgroup after the "other" category (~35–62%), summarizing multiple diseases. Thus, the results from the associated disease analysis and the first approach on PASS that considered the

possible activity of the organoboron compounds as chemosensitizers formed the basis for the evaluation of the boronic acid derivatives as possible drugs for cancer treatment in a model of stage IV breast cancer: 4T1 cells in culture [33].

Breast cancer is the most common malignant disease in women and is among the leading causes of cancer-related death in women worldwide [34]. The 4T1 breast cancer model is a triple-negative ductal carcinoma negative for estrogen receptor (ER), progesterone receptor (PR), and human epidermal growth factor receptor 2 (HER2) expression [33,35]. Interestingly, no specific therapy is available for patients with this cancer subtype, who have a poor prognosis [36].

In the absence of therapeutic targets, chemotherapy plays a vital role in treatment. Doxorubicin (DOX), an anthracycline with anticancer activity, is one of the most effective chemotherapeutic agents against solid tumors, including breast cancer [6]. DOX triggers cell death by apoptosis and necrosis. Even when DOX is effective in cancer treatment, it induces several adverse effects such as myelosuppression, nausea/vomiting, mucositis, diarrhea, and cardiotoxicity that progresses to congestive heart failure [6]. The search for new molecules that can contribute to breast cancer treatment is highly active, seeking greater potency and fewer adverse effects.

As the in silico analysis suggested that the compound's possible effects could be related to cancer, we employed a cell line for which growth does not depend on hormonal stimuli and used DOX as a positive control. As expected, the results showed that boronic acid had no effect on the cell number, being considered a negative control. Unfortunately, compounds **2** and **3** precipitated when solubilized in the cell culture medium, and their effects were not evaluated. The reason for this observed phenomenon was not elucidated, but the relationship with water/lipid solubility is unsupported for the predicted ClogP or LogS values (see Table S2). The cLogP values for compounds **2** and **3** suggest higher hydrophobicity than for the other three tested BCC; assays using nonpolar solutions as vehicles could improve solubility, but this was beyond the scope of the current study.

The analysis of the effects of compound **1** ($IC_{50}$ = 0.1969 μM) and compound **4** ($IC_{50}$ = 0.2251 μM) showed that both compounds induced cytotoxic and perhaps anti-neoplastic effects. In the structure of both compounds, a conjugate system (i.e., naphthalene and phenanthrene moieties) is present, suggesting that this molecular feature is relevant for the induction of antiproliferative effects, at least in the model used, and further studies in this regard are warranted. Interestingly, compound **5** induced no effects. Besides boric acid and phenylboronic acid, which are active in cancer cells at supra-micromolar concentrations [24,37], and the active compounds in this study, it is one of the BCC, with a boron atom and at least two hydroxyl groups. Moreover, this molecule shares two benzyl groups, as do compounds **1** and **4**. Still, such six-member cycles are separated by a five-member cycle, which uncouples the resonance effect of the naphthalene moiety, thus reinforcing the proposal.

The effects of DOX were more potent ($IC_{50}$ = 0.02371 μM) as compared to those of compounds **1** ($IC_{50}$ = 0.1969 μM) and **4** ($IC_{50}$ = 0.2251 μM). However, their effects were induced in the sub-micromolar range and have the potential for more detailed studies. These results warrant further studies to determine the pathways involved in the induced effects, as well as the inclusion of other structurally related compounds.

Although studies including similar molecules are scarce, our results agree with the reported results of F. Zhou et al. exploring the cytotoxic effects of phenanthrene derivatives from extracts of Bletilla striata against A549 lung cancer cells ($IC_{50}$ < 10 μM) [38]. These authors found that complex molecules such as bi-phenanthrenes induced stronger effects than simple phenanthrenes, with cytotoxic effects at levels higher than 100 μM. This indicates that the presence of boronic acid in simple phenanthrenes can increase their potency, as our results also suggest.

Our results also agree with the results reported by E. Spaczynska et al. on the anticancer effects of naphthalene derivatives ($IC_{50}$ > 10 μM) [39]. Additionally, our results could be related to the reports of a dipeptidyl boronate derivative active on triple-negative breast cancer cells [40], albeit that the compound did not have the moieties suggested as key in the

current study (exposed boron-gem-dihydroxyl and two conjugated six-membered cycles). In relationship with our findings, it was also reported that sodium borates (at concentrations higher than 500 μg/mL) have antiproliferative effects in breast cancer cells. Although the involved mechanism for this cytotoxicity is unclear, the results suggested the involvement of the PD-1/PD-L1 signaling pathway and cytokine modulation [41]. In addition, boronic acids have been proposed to act as arginase inhibitors and potent immunomodulators of the response to many types of cancer [42].

Our results in skeletal muscle myoblasts, a noncancer cell culture, are relevant since we found no cytotoxic effects of compounds **1** and **4**, suggesting that the effects of these molecules are exerted only in cancerous cells, and this may decrease chemotherapeutic toxicity in normal cells.

## 4. Materials and Methods

### 4.1. Compound Selection

Five boronic acid derivatives (Figure 1) were selected based on the main molecular features relevant to the protein–ligand interaction establishment of bioactive compounds [43–45], mainly a molecular weight lower than 500 g/mol, the inclusion of an aromatic moiety, and the presence of electro-donating/attracting atoms [46].

### 4.2. Prediction of Probable Biological Activity and Therapeutic Target Screening

As a first approach to searching for the probable biological activity of the selected compounds, an analysis in the Prediction of Activity Spectra for Substances (PASS) online server was performed (http://www.way2drug.com/passonline/, accessed on 10 October 2022) [14]. After that, three reverse-screening online servers were used to explore the most probable therapeutic targets for each compound; the SMILES identifier for each compound was used to start the calculation based on the derived structure for SuperPred (https://prediction.charite.de/subpages/target_prediction.php, accessed on 29 March 2022) [20], PASS-Targets (http://www.way2drug.com/passtargets/, accessed on 29 March 2022) [22], and Polypharmacology browser 2 (PPB2) (https://ppb2.gdb.tools/, accessed on 29 March 2022) [21], using the Extended Connectivity fingerprint ECfp4 NN(ECfp4) + NB(ECfp4) method for the latter.

Finally, common targets among compounds and among different servers were used to find associated diseases wherein the proteins are involved using the MalaCards Human Disease Database (https://www.malacards.org/, accessed on 29 March 2022) [23].

### 4.3. Pharmacological Assessment

Dose–response curves were developed for each compound. We evaluated cell viability using a model of the triple-negative type of breast cancer in vitro—4T1 cells in culture. Additionally, we evaluated the effect of the compounds in a non-tumor cell line (C2C12, mouse myoblasts). Doxorubicin [0.00001–10 μM] was used as a positive control, while boric acid [0.2–1.2 μM] was used as a negative control.

### 4.4. Cell Culture

Type 4T1 cells (ATCC CRL-2539) were cultured under standard conditions, using RPMI-1640 medium with 10% fetal bovine serum and 1% antibiotic, with a controlled temperature and atmosphere at 37 °C and 5% of $CO_2$, respectively. Cells were grown in a monolayer to 80% completeness.

C2C12 cells (ATCC CRL-1772) were cultured using DMEM medium under the same standard conditions.

### 4.5. Estimation of the Number of Cells by Crystal Violet Assay

A total of $5 \times 10^4$ 4T1 cells were seeded in 96-well plates. The next day, the cells were treated with different concentrations of the compounds for 5 days under standard culture conditions, exchanging the culture medium on Day 3. At the end of the 5 days, the cells were washed with $1\times$ HBSS and incubated with 50 μL of 0.5% crystal violet staining

solution for 20 min at room temperature on a bench rocker. After washing three times with distilled water, the plate was inverted to remove any remaining liquid. The plate was air-dried, 200 μL of glacial acetic acid 0.2M was added, and the plate was incubated for 20 min at room temperature. The optical density of each well was obtained at 570 nm (OD 570) in a plate reader. Viability was expressed as the percentage of that for the control group.

*4.6. Statistics*

All assays were performed in triplicate in at least two independent assays. The results are presented as the mean ± SEM. Statistical analysis and IC50 determinations were performed using GraphPad Prism version 9.0 software.

## 5. Conclusions

In the present work, by using in silico and in vitro approaches, we demonstrated the relationship between the potential anticancer activity of two boronic acid derivatives (6-hydroxynaphthalen-2-yl boronic acid (**1**) and phenanthren-9-yl boronic acid (**4**)) suggested by reverse screening and their antiproliferative effect on breast cancer cells. However, further study is required to explore/find the mechanisms involved in the cytotoxic effects of these molecules and to evaluate the relationship of the found effects with interaction on the predicted or suggested targets.

## 6. Perspectives

In recent years, advances in antitumor treatment have modified the prognosis of patients affected by a wide variety of neoplasms, increasing survival and even complete cure of the disease. In the case of tumors such as breast cancer or lymphoblastic leukemia, cure rates have increased considerably. However, therapeutic success and the consequent increase in survival have also increased the frequency of late complications.

Some adverse effects depend on toxicity in tissues unrelated to the tumor. A clear example is the cardiotoxicity induced by anthracyclines such as doxorubicin, which can produce cardiomyopathy and heart failure in many subjects, even after several years.

The administered dose of doxorubicin is directly related to the appearance of complications. The damage is related to the mechanism of action, which depends on its interaction with DNA and the inhibition of topoisomerase II, the generation of reactive oxygen species, and further damage to DNA. All these mentioned mechanisms damage tumor cells and normal cells, particularly those that are more susceptible, such as skeletal or cardiac muscle.

Consequently, the search for drugs capable of limiting tumor growth without affecting other susceptible cells is an active research point worldwide.

The results reported in this work using in silico and in vitro methodologies allow us to suggest with a high probability that the boron derivatives 6-hydroxynaphthalen-2-yl boronic acid (**1**) and phenanthren-9-yl boronic acid (**4**) could be potential candidates for consideration as antineoplastic drugs with selectivity towards tumor cells without affecting normal cells. The results open different avenues of study, requiring an investigation into the effects of these boronic derivatives in solid tumors of various origins and in combination with traditional antineoplastics, seeking to limit adverse effects and increase antitumor effectiveness.

However, the need to implement new studies to characterize these compounds' effects in depth is clear.

**Supplementary Materials:** The following supporting information can be downloaded at: https://www.mdpi.com/article/10.3390/inorganics11040165/s1, Table S1. Names of protein targets determined from UniprotKB server. Table S2. cLogP and LogS of tested compounds.

**Author Contributions:** All authors participated in the conception of the study and in the experimental design and data analysis. M.O.-F., G.C. and N.N. wrote the first draft. All authors revised and edited the manuscript. All authors have read and agreed to the published version of the manuscript.

**Funding:** This research was funded by the National National Institute of Perinatology, grant no. 2018-1-157 and Instituto Politéc-nico Nacional grant no. 20220812. The authors would like to thank

the Secretaría de *Investigación y Posgrado del Instituto Politécnico Nacional* for their financial support of this project.

**Institutional Review Board Statement:** Not applicable.

**Informed Consent Statement:** Not applicable.

**Data Availability Statement:** Not applicable.

**Conflicts of Interest:** Dr. Ceballos is a stockholder of Epirium, Inc. All other authors declare no conflict of interest.

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
