# Peer review of "Reverse Screening of Boronic Acid Derivatives: Analysis of Potential Antiproliferative Effects on a Triple-Negative Breast Cancer Model In Vitro"

_inorganics, doi:10.3390/inorganics11040165_

Round 1

Reviewer 1 Report

The study conducted by Ortiz-Flores et al. on the “Reverse screening of boronic acid derivatives: Analysis of potential antiproliferative effects on a triple-negative breast cancer model in vitro” aims at presenting a target for a triple-negative type of breast cancer. As

Abstract

1.       Remove the main objective of this work

2.       Remove methodology

3.       Remove results

4.       Remove In conclusion

Introduction

1.       In this sense, these characteristics point to boronic acid derivates as compounds with potential therapeutic applications

Can be rewritten as: In this sense, the aforementioned characteristics of boron point to boronic acid derivates as compounds with potential therapeutic applications  

2.       In this sense, the main objective of this work was to …….

Simply say, In this study, the main objective was to ……….

3.       The introduction of this work lacks in-depth background on the study context. It is poorly referenced, with a lot of grammatical blunders.

Materials and method

1.       It is careless negligence for the authors to copy and paste into their manuscript from the internet. The authors should be able to type http://www.way2drug.com/passonline/

2.       What is, “Was used to start calculation based by derived structure?”

Results and discussion

1.       There is no reference to Figure 1.

2.       What is the importance of Figure 2?

3.       The authors’ report on the results and discussion is not clear. Therefore, for clarity of their investigation, authors are advised to

a.       Sectionalize the various online tools used, and state why each is important.

b.       Present clearer Figures and appropriately explain the contents in the Figures

In general, this manuscript requires thorough editing and logical presentation; its present form is not publishable. The introduction must be worked upon, and the methodology, results, and discussion must be clearer.

Author Response

Inorganics-2326122R1

Reviewer #1:

R: Thank you for your comments, we believed that all your consideration have been fulfilled.

Abstract

  1. Remove the main objective of this work
  2. Remove methodology
  3. Remove results
  4. Remove In conclusion

R: Thank you for your comments. All changes were done in the abstract, as you requested.

Introduction

  1. In this sense, these characteristics point to boronic acid derivates as compounds with potential therapeutic applications

Can be rewritten as: In this sense, the aforementioned characteristics of boron point to boronic acid derivates as compounds with potential therapeutic applications  

  1. In this sense, the main objective of this work was to …….

Simply say, In this study, the main objective was to ……….

  1. The introduction of this work lacks in-depth background on the study context. It is poorly referenced, with a lot of grammatical blunders.

R: Thank you for your comments. The introduction section was edited considering all your comments, and references (+20) were added, and the entire manuscript has been proofread.

Materials and method

  1. It is careless negligence for the authors to copy and paste into their manuscript from the internet. The authors should be able to type http://www.way2drug.com/passonline/

R: Thank you for your comments. We apologize for the mistake. We have edited the text following your suggestion. Some links had been automatically generated in the recent versions of Word app.

However, we do not see the difference

  1. What is, “Was used to start calculation based by derived structure?”

R: Thank you for your comments. It was edited for clarity.

Results and discussion

  1. There is no reference to Figure 1.

R: Thank you for your comments, we are sorry for the mistake, Figure 1 is now referred in the text.

  1. What is the importance of Figure 2?

R: Thank you for your comments. Each Figure presents the results from each server. We have highlighted some data in the results section. We believed that to compare results from servers all figures must show the results obtained.

  1. The authors’ report on the results and discussion is not clear. Therefore, for clarity of their investigation, authors are advised to
  2. Sectionalize the various online tools used, and state why each is important.
  3. Present clearer Figures and appropriately explain the contents in the Figures

R: Thank you for your comments. We have added an explanation of what is observed in each figure created from in silico tools.

Also, we have extended the comparison with other boron-containing compounds active in cancer cells.

In general, this manuscript requires thorough editing and logical presentation; its present form is not publishable. The introduction must be worked upon, and the methodology, results, and discussion must be clearer.

  1. Thank you for your comments. The manuscript has been extensively edited for clarity and for language mistakes.

Author Response

Inorganics-2326122R1

Reviewer #2

The manuscript submitted presents an interesting and novedous study on boronic acids as potential anticancer drugs. The reverse screening methodology used is attracting a great deal of interest as well as the design of novel boron-containing small molecules in new drugs discovery. The methodology used is the appropriate one for this type of research, the results are interesting and novedous and may attract the interest of researchers in this topic. In general the quality and relevance of the studies and results may fit the standard required for their publication as an article in Inorganics.

R: Thank you for all your comments. All of them enriched our work and all were taken into account to improve the presentation of the manuscript.

However, I would tend to suggest a major revision in order to ease the reading and understanding and to make the manuscript more attractive. In the following paragraphs I will detail some points that should be considered by the authors as well as some suggestions

  1. In my opinion this manuscript is difficult to read. For instance, the “excessive use of the compounds’names” in the main text and also in some legends or captions, and the huge number of acronims, abbreviations, etc… used makes its reading extremely tough. So I would suggest to introduce some changes so that the reader could know easily the meaning of the acronims and abbreviations (maybe a list could be incorporated somewhere in the manuscript, or as supplementary information) and also identify the 5 compounds at a first glance, this for sure will ease the reading.

R: Thank you for all your comments. The entire manuscript was revised and edited to improve the presentation. We believe that acronyms and reference to compounds are now clearer.

Table 1 should be deleted and replaced by a new Figure 1, where the authors could present the chemical formulae of the 5 compounds. The two compounds labeled as 1 and 2 in the current version could be shown in the first line and those with the tri- (compound 4) or tetracyclic (compounds 3 and 5) units (in the following line). If this is to be done, I would suggest to use the same orientation of the “B(OH)2” array in the five chemical formulae and maybe the use of colors for the different atoms or even for the R group bound to the “B(OH)2” units in all cases could make the presentation of the compounds more attractive than in a Tabular form. The IUPAC’s name of the corresponding compounds could also be included below each formula and the identification codes could be presented in bold character in the same plot. Obviously, is this is done, all the following Figures and Tables should be renumbered accordingly.

If the presentation of the compounds is clear, the names of the compounds could be replaced by their identification codes in bold along the text, captions, etc... See for instance captions of Figures 2, 3 where the last lines could be deleted and then, “for compounds 1-5” could be added after “…. server”.

R: Thank you for all your comments. We agree with your suggestion, and we have edited it. The compounds are now referred as you suggested. Thank you!

  1. Abstract: a) Please add the identification codes for the compounds selected for this study in parenthesis and in bold (if possible, later on I will explain why I suggest this change) after their IUPACs’ names and then replace the names of the compounds mentioned in line 3 (from the bottom of this paragraph) by their corresponding identification codes, b) The words “Methodology” (line 6) and “Results” (line 10) and “ In conclusion” (line13) should be removed and, c) here and, also in other parts of the manuscript, please unify “ in vitro” and “in silico” so that they appear in itallics in all paragraphs.

R: Thank you for all your comments. We have edited the abstract to be appropriateness.

  1. Introduction: Is it possible to add a new figure with the chemical formulae of the 3 boronic- acid based drugs approved by FAD: Bortezomib, Ixazonib (last line of p. 1) and Vaborbactam (top of p. 2)?

R: Thank you for all your comments We have not added the figure because these and other FDA-BCC approved for human use are not structurally-related to our tested compounds.

  1. Sections: “3.Results and 4. Discussion” In my opinion, these parts of the manuscript should be reorganized in order to ease the reader’s understanding. For instance, in section 4 (of the 2 submitted manuscript), the authors quote quite often Figures presented in section 3. The reader “gets a bit dizzy” and I would recommend to write down a new section entitled “Results and 2 Discussion” and to reorganize the information presented. For instance, the authors could move the first ca. 3 lines just below the heading “4. Discussion” to page 6, as an introduction to section 3 or subsection 3.1. Then they could present the results of “ 3.1. Prediction of probable biological activity, as they are in the current version and afterwards the text actually presented in the first paragraph of “section 4. Discussion”. Later on they could use a similar strategy with section 3.2. Therapeutic targets prediction, that is to say presenting first the results and afterwards in the same section the results, this could be applied to all subsections. In this way the figures quoted while discussing the results would be quite close to the text where the authors mention, for instance, the common targets (see page 10, 2nd paragraph).

R: Thank you for all your comments. The sections were no merged. However, we increased the details in the results section. And the discussion was enriched with the comparison of the action of other BCC active in neoplastic cells, also related to the mechanisms of action and common targets.

Besides these,please re-write:

  1. i) the paragraphs on page 8 in a shorter a clearer way,
  2. ii) the sentence “they are in the range “ in page 12, do the authors mean in the “ nanomolar range”?? and

R: Thank you for all your comments. We have edited it as ‘sub-micromolar range’ as the activity was observed in decimals of micromolar concentration.

iii) those of page 12 just above the one starting as “ In conclusion…”.

I’m pretty sure that these changes would make the paper clearer, and more ease to read and understand than the current version.

R: Thank you for all your comments Sincerely thanks to you for the suggestion to improve the presentation in each section.

In some parts of these sections the IC50 values are given with 4 decimal digits. I have the feeling that the accuracy of the method used to determine these values do not allow to give so many significant digits.

R: Thank you for all your comments. The Prism version # 9 used for the analysis provides a such number of decimals, we copy the number as provided

  1. In addition to the comments concerning the “transformation of the old Table 1, into a new Figure and the edition of an additional new Figure for the Introduction section, I would like to make a few comments on several figures:
  2. a) Figure 4, in my opinion is not too clear. The identification codes of the potential targets are too small they should be enlarged; letters A, B, C, D and E refer to the plots but these letter could be replaced by the corresponding number assigned to each compound. Figure 5 has the same problem.

R: Thank you for all your comments. We have edited for homologous identification of compounds. All of them are compound 1 to 5 in the text.

  1. b) Figure 6: is it possible to show the two representations in the same graph?, maybe the use of two different colors could be useful. Something similar could be carried out with different color will help, Fig. 8? And possibly the 3 curves of Figure 7 could also be presented in a single graph (using different colors), in this case probably the size of the plot should be enlarged compared with those

shown currently in this figure.

R: We have edited the figures we hope that they are clearer in the present form.

Moreover, in my opinion the sizes of the characters and codes presented in Table 2 are too big.

R: We have changed the size of characters.

  1. Bibliography, I have read carefully latest papers published in Inorganics and in all of them citations of the references in the manuscript appear in brackets (i.e. [10], [12-15, 18], etc…) instead of parenthesis. In the current version of this manuscript (2) may refer to reference n.2 or to compound 2, both appear as numbers in parenthesis. Moreover, in the citation list commonly the names of the author appear as follows: Author n.1, initial(s); Author n. 2, initial(s); etc.., please use this format of presentation in the References section.

See also the presentation format for the abbreviations used for the journals’ names, i.e. Ref. 2: Aust. J. Chem. The “dots” ( = “.”) are missing in most of the references of the submitted version. Commonly bold type is used for the year.

R: Thank you for all your comments. We have edited the references in agreement with the guidelines.

In page 9, just below Figure 7:

Authors mention that “ compounds 2 and 3 were not evaluated since these molecules tend to precipitate” , I wonder whether they tried to estimate the log S for these compounds, as a first approach and just to fulfil my curiosity, I used the Chem Draw program to estimate the “log S” values for the 5 compounds and the values obtained for 2 and 3 are clearly more negative ( log S ca. -4.11 for 2 and -4.0 for 3) than those obtained for 1 ( ca. -2.53 ), 4 ( ca. -2.26 ) and 5 (ca. -3.4 ). These log S are only estimated values, but probably the experimental ones will follow a similar trend. As far as I know, most of active drugs have log S between 0 and -4).

R: Thank you for all your comments. We have edited these sentences. Moreover, we have calculated LogS and LogP (Table S2) and added some comments.

  1. Additional minor comments:
  2. a) In page 5: Table 2 caption “(see Table S1 for details)” I couldn’t find this Table, was it submitted?
  3. Typographic errors:

some representative examples : - page 3 ( line 3): cultured ( the “d” was missing); line 5 please use the proper symbol for degrees that is to say it should appear as 37 C

- page. 5: Figure 2 right hand side “ No probability” ( b was missing)

heading of the section: please check: “3.34. T1 Viability D/R curves”

- page. 7: heading of the section: please check: “ 3.34. T1 Viability D/R curves

The manuscript contains more typos but, truly it is impossible to list all of them.

R: Thank you for all your comments. We have revised and edited the entire manuscript to avoid typos and grammar mistakes.

In conclusion, I really think that, the study is complete, interesting and the results achieved new and relevant, but they could be presented in a nicer, clearer and more attractive way. As indicated in the previous points the number of actions needed to improve substantially the quality of the presentation and to ease the reading and understanding is high and this is why I recommended a major revision of this manuscript. However, I encourage the authors to do an additional effort to revise the manuscript according to the comments and suggestions indicated above.

R: Thank you for all your comments and suggestions. All of them have been considered to improve the content of our manuscript.

Round 2

Reviewer 1 Report

The methods engaged in this work must be systematically presented; state why using them. Then, proceed to the results of each of the methods. 

Author Response

Inorganics-2326122R

Reviewer #1:

The methods engaged in this work must be systematically presented; state why using them. Then, proceed to the results of each of the methods. 

  1. Thank you for all your precise comments. We followed the instructions for authors that state that results and discussion precede the material and methods section.

However, we add small paragraphs for clarity at the beginning of each result.

Reviewer 2 Report

The new version of this manuscript is undoubtely clearer and results much  more attractive to readers than the previous one.  its reading and understanding is much easier than before.

Despite I suggested additional modifications (i.e. to merge the sections and discussion sections) I understand the reasons given by the authors to justify why they decided to keep them separetely.

On theses bases and after having read carefully  the new version and the cover letter of the authors, now I think that the current version  is suitable for its publication  in this journal, subject to a few minor corrections that probably could be carried out  during the edition of the "proofs" .  In particular,

a) Please check the caption of Fig. 8: "C. Effect of compound 5 (there is no plot C);

b)  in line 154 (please the bold format for the identification codes of the compounds...in  "as compounds 2 and 3;  line 380 has a similar problem but now with " (1)" and "(4), 

c) Please check section "6. Perpectives"   For me, the style and type of lettering in the whole section  (maybe Arial or Courrier)  looks different than that of the remaining and preceeding sections (that were written using a Times New Roman or similar format).

Author Response

Inorganics-2326122R2

Reviewer #2:

The new version of this manuscript is undoubtely clearer and results much  more attractive to readers than the previous one.  its reading and understanding is much easier than before.

Despite I suggested additional modifications (i.e. to merge the sections and discussion sections) I understand the reasons given by the authors to justify why they decided to keep them separetely.

  1. Thank you for all your comments

On theses bases and after having read carefully  the new version and the cover letter of the authors, now I think that the current version  is suitable for its publication  in this journal, subject to a few minor corrections that probably could be carried out  during the edition of the "proofs" .  In particular,

  1. Please check the caption of Fig. 8: " Effect of compound 5 (there is no plot C);
  2. Thank you for all your comments. We are sorry for the mistake, we delete “C”
  3. in line 154 (please the bold format for the identification codes of the compounds...in  "as compounds 2 and 3;  line 380 has a similar problem but now with " (1)" and "(4), 
  4. Thank you for all your comments. Identification numbers are now in bold format

  1. c) Please check section "6. Perpectives"   For me, the style and type of lettering in the whole section  (maybe Arial or Courrier)  looks different than that of the remaining and preceeding sections (that were written using a Times New Roman or similar format
  2. Thank you for all your comments. The template use Palatino Linotype in Perspectives is Palatino, we now unify the lettering, we sorry for the mistake

Round 3

Reviewer 1 Report

OK.